# Extensive Expression of the Virulome Related to Antibiotic Genotyping in Nosocomial Strains of *Klebsiella pneumoniae*

**DOI:** 10.3390/ijms241914754

**Published:** 2023-09-29

**Authors:** Gloria Luz Paniagua-Contreras, Areli Bautista-Cerón, Rosario Morales-Espinosa, Gabriela Delgado, Felipe Vaca-Paniagua, Clara Estela Díaz-Velásquez, Aldo Hugo de la Cruz-Montoya, Luis Rey García-Cortés, María Patricia Sánchez-Yáñez, Eric Monroy-Pérez

**Affiliations:** 1Facultad de Estudios Superiores Iztacala, Universidad Nacional Autónoma de México, Tlalnepantla 54090, Mexico; areli.bautista@outlook.com (A.B.-C.); patysy_2008@hotmail.com (M.P.S.-Y.); 2Departamento de Microbiología y Parasitología, Facultad de Medicina, Universidad Nacional Autónoma de México, 04510, Mexico; marosari@unam.mx (R.M.-E.); delgados@unam.mx (G.D.); 3Unidad de Biomedicina, Facultad de Estudios Superiores Iztacala, Universidad Nacional Autónoma de México, Tlalnepantla 54090, Mexico; felipe.vaca@iztacala.unam.mx (F.V.-P.); cdiazvelasquez@aol.com (C.E.D.-V.); audelacm@gmail.com (A.H.d.l.C.-M.); 4Laboratorio Nacional en Salud, Diagnóstico Molecular y Efecto Ambiental en Enfermedades Crónico-Degenerativas, Facultad de Estudios Superiores Iztacala, Universidad Nacional Autónoma de México, Tlalnepantla 54090, Mexico; 5Subdirección de Investigación Básica, Instituto Nacional de Cancerología, Ciudad de México 14160, Mexico; 6Instituto Mexicano del Seguro Social, Naucalpan de Juárez 53370, Mexico; luis.garciaco@imss.gob.mx

**Keywords:** multidrug-resistant, virulome expression, antimicrobial resistance genotype, pulse field gel electrophoresis

## Abstract

The emergence of hyper-virulent and multidrug-resistant (MDR) strains of *Klebsiella pneumoniae* isolated from patients with hospital- and community-acquired infections is a serious health problem that increases mortality. The molecular analysis of virulome expression related to antimicrobial-resistant genotype and infection type in *K. pneumoniae* strains isolated from patients with hospital- and community-acquired infections has been poorly studied. In this study, we analyzed the overall expression of the virulence genotype associated with the antimicrobial resistance genotype and pulse field gel electrophoresis (PFGE) type (PFtype) in *K. pneumoniae*. We studied 25 strains of *K. pneumoniae* isolated from patients who developed bacteremia and pneumonia during their hospital stay and 125 strains from outpatients who acquired community-acquired infections. Susceptibility to 12 antimicrobials was determined by Kirby–Bauer. The identification of *K. pneumoniae* and antibiotic-resistance genes was performed using polymerase chain reaction (PCR). To promote the expression of the virulence genes of *K. pneumoniae*, an in vitro infection model was used in human epithelial cell lines A549 and A431. Bacterial RNA was extracted with the QIAcube robotic workstation, and reverse transcription to cDNA was performed with the Reverse Transcription QuantiTect kit (Qiagen). The determination of the expression of the virulence genes was performed by real-time PCR. In addition, 57.3% (n = 86) of the strains isolated from patients with hospital- and community-acquired infections were multidrug-resistant (MDR), mainly to beta-lactam antibiotics (CB, AM, CFX, and CF), aminoglycosides (GE), quinolones (CPF and NOF), nitrofurantoin (NF), and sulfamethoxazole/trimethoprim (SXT). The most frequently expressed genes among strains isolated from hospital- and community-acquired infections were adhesion-type, *ycfm* (80%), *mrkD* (51.3%), and *fimH* (30.7%); iron uptake, *irp2* (84%)*, fyuA* (68.7%), *entB* (64.7%), and *irp1* (56.7%); and protectins, *rpmA* (26%), which were related to antibiotic-resistance genes, *bla_TEM_* (96%), *bla_SHV_* (64%), *bla_CITM_* (52.6%), *bla_CTXM-1_* (44.7%), *tetA* (74%), *sul1* (57.3%), *aac(3)-IV* (40.7%), and *aadA1* (36%). The results showed the existence of different patterns of expression of virulome related to the genotype of resistance to antimicrobials and to the PFtypes in the strains of *K. pneumoniae* that cause hospital- and community-acquired infections. These findings are important and may contribute to improving medical treatment strategies against infections caused by *K. pneumoniae*.

## 1. Introduction

The emergence of multidrug-resistant *Klebsiella pneumoniae* strains associated with hospital- and community-acquired infections, such as bacteremia, meningitis, pneumonia, and urinary tract infections (UTIs) [1,2,3], is a major concern for choosing the appropriate medical treatment [4]. The multidrug resistance of *K. pneumoniae* occurs mainly through the horizontal transfer of resistance genes through mobile genetic elements, such as plasmids, transposons, and integrons [5,6]. The frequency of genes encoding extended-spectrum beta–lactamases (ESBL) in *K. pneumoniae* strains has increased [7], as have genes for resistance to aminoglycosides, quinolones, tetracycline, and carbapenemases [8,9,10].

Recently, an increase in multidrug-resistant (MDR) and hyper-virulent (hvKpn) variants of *K. pneumoniae* [11,12], which cause highly invasive infections such as liver abscesses, endophthalmitis, meningitis, and septic arthritis [13], have been described in both healthy and immuno-compromised individuals [3]. The hypervirulence of *K. pneumoniae* has been associated with the hypermucoviscous phenotype, the K1 and K2 capsular serotype, the presence of the *magA* (mucoviscosisty-associated gene A) and *rmpA* (regulator of mucoid phenotype A) genes, and the salmochelin (*iro*) and aerobactin (*iuc*) siderophore systems [14].

The pathogenic capacity of *K. pneumoniae* is associated with the expression of a large number of virulence markers, which include adhesins (fimbriae Type 1 and Type 3), iron-acquisition systems, protectins, and toxins [15]. The genes that encode the fimbriae (*fimH-1*, *kpn*, *ycfM*, and *mrkD*) are considered the main virulence markers of *K. pneumoniae* that mediate adhesion to host cells. Therefore, their expression favors bacterial colonization [16], while the expression of siderophore genes, such as enterobactin (*entB*), yersiniabactin (*irp1*, *irp2*, *fyuA*, and *ybtS*), salmochelin (*iroN*) and aerobactin (*iutA*), allowing for iron uptake and facilitating survival and bacterial multiplication during infections [17], while the expression of the *cnf-1* and *hlyA* toxin genes promote the degradation of the host tissues, facilitating bacterial dissemination [18].

The increase in mortality rates in patients with hospital- and community-acquired infections caused by hyper-virulent and multidrug-resistant strains of *K. pneumoniae* is a critical health situation [19], so it is currently important to analyze the properties molecular virulence and antimicrobial resistance of the new variants of *K. pneumoniae* with the purpose of improving medical treatment alternatives.

Molecular characterization of hyper-virulent hospital strains of MDR *K. pneumoniae* in Mexico [20,21], and in other parts of the world, have been studied [22,23,24]. However, reports on the expression of virulence markers, and their correlation with multidrug-resistance and clinical origin, are scarce. Therefore, in this study, we aimed to establish an in vitro model of infection using human cell lines to globally and comprehensively determine the virulome expression related to the antimicrobial resistance genotype and pulse field gel electrophoresis (PFGE) type in a group of *K. pneumoniae* strains from different infections.

## 2. Results

### 2.1. Multidrug-Resistance Phenotype

Utilizing the Kirby–Bauer disk diffusion method, our analysis revealed notable levels of antibiotic resistance among the tested strains, with the highest percentages of resistance observed for carbenicillin, ampicillin, and cefotaxime, as well as for nitrofurantoin (Table 1). The 57.3% (n = 86) of the *K. pneumoniae* strains isolated from patients with hospital- and community-acquired infections were multidrug-resistant in the range of 4–12 antimicrobials, within which more than 40% (n = 60) of the strains presented the phenotype of multi-resistance to nine antibiotics from different groups; beta-lactams (CB, AM, CFX, and CF), aminoglycosides (NET), quinolones (CPF and NOF), nitrofurans (NF), and sulfonamide/trimethoprim (SXT).

### 2.2. Origin of the Strains and In Vitro Infection of Human Epithelial Cell Lines to Determine Viruloma Expression

The majority of hospital-acquired *K. pneumoniae* strains were isolated from patients with bacteremia (21/150; Table 2) and community-acquired infections from patients with UTIs (61/150) and respiratory infection (53/150). Using the real-time PCR method, we found that the most frequently expressed genes among the strains after the in vitro infection model of the human epithelial cell lines A549 and A431 were the genes encoding adhesins (*ycfm*, *mrkD*, and *fimH*), the genes encoding iron-acquisition systems (*irp2*, *fyuA*, *entB*, and *irp-1*) and protectins (*rpmA*; Table 2). The frequency of expression of most virulence genes in the strains according to clinical origin was very similar, except for adhesin (*mrkD*) and enterobactin (*entB*), whose percentages were higher in pneumonia and bacteremia (hospital-acquired infections), respectively, while yersiniabactin (*ybtS*) and protectin (*rpmA*) percentages were higher in others (community-acquired infections). The genes expressed with the least frequency after the in vitro model of infection of the cell lines with *K. pneumoniae* strains were *hlyA* (n = 6) and *magA* (n = 1), both from strains isolated from patients with infections acquired in the community. The gene that codes for the CNF-1 toxin was not expressed in any of the strains studied.

### 2.3. Detection of Antibiotic Resistance Genes by Conventional PCR

The most frequently detected antibiotic-resistance genes by the conventional PCR method among *K. pneumoniae* strains were beta-lactams (*bla_TEM_*, *bla_SHV_*, *bla_CITM_*, and *bla_CTXM-1_*), tetracycline (*tetA*), sulfonamide (*sul1*), gentamicin (*aac(3)-IV*), streptomycin (*aadA1*), and trimethoprim (*dfrA1*). No statistically significant differences were found between the frequency of most antibiotic-resistance genes in strains according to clinical origin, except for *bla_CTX-1_* and *aac(3)-IV*, where frequencies were higher in strains isolated from bacteremia and pneumonia of patients with hospital acquired infections than in others (Table 3), respectively. The antibiotic resistance genes *bla_CTXM-2_* (n = 4) and *qnr* (n = 2) in strains isolated from patients with hospital-acquired infections (UTI and respiratory infection) had the lowest frequency.

### 2.4. Virulence Genes Expression According to Antibiotic Resistance Genes

Overall, in the *K. pneumoniae* strains isolated from patients with hospital-acquired infections, our real-time PCR assays showed an elevated expression of adhesion genes (*mrkD* and *ycfM*) and iron-acquisition systems (*entB*, *irp1*, *irp2*, and *fyuA*) associated with the presence of antibiotic-resistance genes to beta-lactam antibiotics (*bla_SHV_*, *bla_CITM_*, *bla_TEM_*, and *bla_CTXM-1_*), sulfonamides (*sul1*), and tetracycline (*tetA*) (Table 4). However, statistically, the individual frequency of expression of each virulence gene related to the detection of each of the antibiotic-resistance genes in the strains was different (Table 4).

Similarly, among the *K. pneumoniae* strains isolated from patients with community-acquired infections (Table 5), we detected a high expression by real-time PCR of adhesion genes (*mrkD* and *ycfM*), and iron acquisition systems (*entB*, *irp1*, *irp2*, and *fyuA*) related to detection by conventional PCR of resistance genes to beta–lactam antibiotics (*bla_SHV_* and *bla_TEM_*) and tetracycline (*tetA*). However, statistically, the individual frequency of expression by real-time PCR of each virulence gene related to the detection by conventional PCR of each of the antibiotic-resistance genes in the strains was different (Table 5).

### 2.5. PFGE and Expression Patterns of Virulence Genes

The PFGE analysis using *XbaI* showed two main clades (A and B) (Figure 1). Clade A, comprising 92 strains, was divided into two compact subgroups, A1 (n = 41) and A2 (n = 51). Subgroup A1 showed a Dice similarity index > 61.1% among the strains, and Subgroup A2 showed a similarity index of 65.1%. Subgroup A1 showed 10 different PFtypes (pulse field types), where the PFtype comprises four strains (41, 48, 84, and 154) isolated from patients with community infections (UTI (n = 2) and respiratory infection (n = 1)), and hospital infections (pneumonia (n = 1)) showed different expression profiles of virulence genes associated with the antibiotic resistance genotype, with strain 84 showing the broadest profile (*mrkD/ycfM/rpmA/entB/irp-1/irp-2/fyuA/iutA/hlyA/aac(3)-IV/bla_SHV_/tetA/bla_CTXM-1_/bla_TEM_*). Similarly, in Subgroup A2, a PFtype comprising nine strains (12, 61, 73, 76, 94, 101, 112, 124, and 133) of different origins was identified, all of which had different virulence gene-expression profiles associated with the antibiotic-resistance genotype.

Clade B, comprising 57 strains, was distributed into two subgroups: B1 (n = 6) and B2 (n = 51) (Figure 1). Subgroup B1 showed a Dice similarity index > 65% among strains, and Subgroup B2 showed a similarity index of 67.9%. Subgroup B1 had only one PFtype, comprising 4 strains (80, 146, 102, and 81) of different origins and with different expression profiles of virulence genes associated with the antibiotic-resistance genotype. In Subgroup B2, 12 different PFtypes were identified, some of which comprised paired strains (119–116; 1–4; and 54–150), and one PFtype that comprised eight strains (50, 72, 122, 140, 142, 149, 151, and 153), all from community infections (UTI = 5; respiratory infection = 2, and other = 1), and with different virulence gene expression profile associated to the antibiotic resistance genotype.

### 2.6. Unsupervised Hierarchical Clustering

Unsupervised hierarchical clustering analysis showed three main groups based on similarities between *K. pneumoniae* strains (Figure 2). Group 1 (strain range 40–108) had the highest number of expressed virulence genes and was characterized by the simultaneous expression of one-to-seven virulence genes. Group 2 (strains between 70–147) had an intermediate number of expressed virulence genes (four to nine genes), and Group 3 (strains between 138–48) had the lowest number and was defined by the expression of 3–10 virulence genes.

Based on the expression of the virulome (VE) by real-time PCR, two different clades were identified: (A) Highly expressed among the strains (*ymfM*, *irp-2*, *fyuA*, *entB*, *irp-1*, and *mrkD*), and (B) Lowly expressed between strains (*fimH*, *iutA*, *rpmA*, *kpn*, *ybtS*, *iron*, *hlyA*, *magA*, and *cnf*). The antibiotic-resistance genotype (ARG) was grouped into two main clades according to their frequency of detection by conventional PCR. Clade C had a large number of genes detected (*bla_TEM_*, *tet(A)*, *sul-1*, *bla_SHV_*, *bla_CTXM-1_*, *bla_CITM_*, *aadA1*, and *aac(3)-IV*), and Clade D was composed by a lower number of gene (*dfrA-1*, *cmlA*, *cat1*, *tet(B)*, *bla_CTXM-_9*, *bla_CTXM-_2*, and *qnr*).

The antibiotic-resistance phenotype (ARP) was characterized by a high frequency of resistance to beta–lactams (AM, CB, CF, and CFX), as well as NF, SXT, and GE. The percentages of the resistance phenotype to beta–lactam antibiotics, aminoglycosides, SXT, and CL in the strains coincided with the resistance genotype for beta-lactams (*bla_SHV_*, *bla_TEM_*, *bla_CITM_*, and *bla_CTXM-1_*), aminoglycosides (*aac(3)-IV)*), sulfonamide (*sul-1*), trimethoprim (*dfrA-1*), and chloramphenicol (*cat1* and *cmlA*) (Figure 2).

Overall, the unsupervised hierarchical analysis showed in the three groups (1, 2, and 3) strains with the same pattern of virolome expression. For example, in Group 2, it is observed in the cladogram that Strains 101, 98, 90, 25, 9, and 21 presented the same expression profile of the virulence genes (*ycfM*, *irp-1*, *fyuA*, *entB*, *irp-1*, and *mrkD*) associated with the frequency of the genotype and phenotype of resistance to antibiotics *bla_TEM_-TET(A)-AM-CB*, and with the clinical origin. Interestingly, this shared pattern of expression was independent of the strain origin because Strains 101, 98, 90, and 9 were isolated from bacteremia of patients with hospital-acquired infections, as well as Strains 25 (infected ulcer) and 21 (respiratory infection) from patients with community-acquired infections. The Strains 113-50-79; 45-7-43 from Group 1, and 102-134, 42-104, and 2-12 from Group 3, also had the same virulome expression pattern.

## 3. Discussion

The emergence of hypervirulent and MDR hospital- and community-acquired strains of *K. pneumoniae* is a serious concern that increases mortality, mainly in immune-compromised patients [14,25]. Numerous virulence factors contribute to the pathogenicity of *K. pneumoniae*, thus increasing the severity of hospital and community infections [26]. Our working group previously determined the frequency of virulence genes by conventional PCR in these *K. pneumoniae* strains (n = 150), as well as its relationship with the hypermucoviscosity phenotype, capsular serotypes, biofilm formation, and the resistance phenotype to antibiotics [27]. In this work, we continue our research by characterizing in these *K. pneumoniae* strains the expression of the virulome by real-time PCR related to the antimicrobial resistance genotype, to the pulse field gel electrophoresis (PFGE) type, and their clinical origin.

In this study, the overall percentages for the expression of different virulence genes in *K. pneumoniae* strains isolated from patients with hospital- and community-acquired infections following the in vitro infection of human epithelial cell lines were very similar, except for the frequency of expression of *mrkD*, *entB*, *ybtS*, and *rmpA* genes, where there were statistically significant differences. We found a higher expression of the adhesion marker *mrkD* in strains from hospital-acquired infections (pneumonia) than in strains from community-acquired infections. The adhesion gene *ycfM* was expressed in all strains isolated from bacteremia (21/21), pneumonia (4/4), and stool culture and tumor biopsy (3/3; other). The co-expression of the adhesion genes *fimH*, *mrkD*, and *ycfM* detected in some strains of *K. pneumoniae*, following the infection of A549 and A431 cell lines, suggests that this expression combination could increase bacterial colonization and persistence in infected tissues. Type 1 fimbriae (*fimH*) was previously found to be frequently expressed during the infection of urinary tissue and participate in the invasion of bladder cells [16], while Type 3 fimbriae (*mrkD*) was found to promote biofilm formation and, consequently, the evasion of the host immune response [28].

Iron is an essential element for electron transport, DNA synthesis, and peroxide reduction, favoring bacterial survival and multiplication during infections [29]. Among the expression of iron uptake genes, *entB* expression was higher in bacteremia and pneumonia (hospital-acquired infections), and in respiratory infection (community-acquired infection) as well. High expression percentages of *irp1* (bacteremia, respiratory infection, and others), *irp2*, and *fyuA* (bacteremia, UTI, respiratory infection, and others) were also found. The simultaneous high expression of the iron uptake genes *entB*, *irp1*, *irp2*, *ybtS*, and *fyuA* by some of the strains after infection of A549 and A431 cell lines highlights the virulence of the strains to cause acute infections because *entB* (enterobactin) and *irp-1* (yersiniabactin) expression increases biofilm formation in *Klebsiella pneumoniae*, causing liver abscess [30]; *ybtS* (yersiniabactin) promotes respiratory tract infection [31], while *irp* and *fyuA* (yersiniabactin) reduce the bactericidal capacity of innate immune cells [32].

The expression of the *rmpA* gene (protectin) was higher in strains from community-acquired infection [respiratory infection (23/53) and others (2/3)] than in that from hospital-acquired infections. The toxin gene *hlyA* (hemolysin A) was only expressed in strains isolated from community-acquired infection [UTI (4/61) and respiratory infection (2/53)]. The expression of the mucoid phenotype A (*rmpA*) gene was detected mainly in the strains from respiratory infection and UTI. This proves the ability of the strains to cause more acute infections because *rmpA* has been previously associated with hvKpn strains, causing liver abscesses and bacteremia [33], while *hlyA*, which was more frequently expressed in strains originating from UTI (community-acquired infection), favors bladder inflammation during UTIs [34].

More than half of the *K. pneumoniae strains* (n = 86) isolated from patients with hospital- and community-acquired infections were multidrug-resistant (MDR), mainly to beta-lactam antibiotics (CB, AM, CFX and CF), aminoglycosides (GE), quinolones (CPF and NOF), nitrofurantoin (NF), and sulfamethoxazole/trimethoprim (SXT), which coincides with the multidrug-resistance described in strains of *K. pneumoniae* isolated from patients with hospital-acquired infections in Mexico [20].

The frequency of the detection of antibiotic-resistance genes in *K. pneumoniae* strains isolated from patients with hospital- and community-acquired infections was very similar, except for *bla_CTXM-1_* (beta-lactam) and *aac(3)-IV* (gentamicin), where the frequency was higher in strains isolated from bacteremia and pneumonia, respectively, both from hospital-acquired infections. High percentages for beta–lactam (*bla_SHV_* and *bla_TEM_*), tetracycline (*tet(A)*), and sulfonamide (*sul1*)-resistance genes were found in strains isolated from patients with hospital-acquired infections (bacteremia and pneumonia) and community-acquired infections (UTI, respiratory infection, infected ulcer, and others). High percentages for the streptomycin- (*aadA1*) and gentamicin- (*aac(3)-IV*) resistance gene were also detected in strains from patients with hospital-acquired infections (pneumonia) and community-acquired infections (UTI, respiratory infection, and infected ulcer). The overall number of *bla_TEM_* (144/150) and *bla_CXTX-9_* (10/150) coincide with those described in *K. pneumoniae* strains isolated from Mexican patients in a health facility [35], while the numbers of *bla_SHV_* (96/150), *bla_CTXM-1_* (67/150), and *bla_TEM_* (144/150) are higher than those described in *K. pneumoniae* strains isolated from hospital-acquired infections in other parts of the world [36,37]. The wide distribution of the resistance genotype to other non-beta–lactam antibiotics, such as tetracycline (*tet(A)*), sulfonamide (*sul1*), gentamicin (*aac(3)-IV*), and streptomycin (*aadA1*) detected in *K. pneumoniae* strains from patients with hospital- and community-acquired infections coincides with the high MDR phenotype found in Mexico in hospital strains of *K. pneumoniae* from bacteremia, UTIs, respiratory infection, and other infections [35,38].

In this study, the correlation of the expression percentages of adhesion genes (*mrkD* and *ycfM*) and iron-acquisition systems (*entB*, *irp-1*, *irp2*, and *fyuA*) with the genotype of resistance to beta–lactam antibiotics (*bla_SHV_, bla_TEM_*, and *bla_CTXM-1_*), sulfonamide, and tetracycline was higher in *K. pneumoniae* strains isolated from patients with hospital-acquired infections compared with strains from community-acquired infections. These results are substantial and suggest that the expression of the virulence genotype related to the antimicrobial-resistance genotype in strains from bacteremia and pneumonia (hospital-acquired infections) may cause more acute infections, increasing patient mortality. Mortality in patients with hypervirulent *K. pneumoniae* pneumonia has been found to be 23.1% [39], 54.3% in bacteremia, 48.9% in intensive care, 43.1% in organ transplantation, and 13.5% in urinary tract infection [40]. The prevalence of the *cnf-1* gene (n = 22) and the *magA* gene (n = 25), which had been previously identified by our research group within the genome of these *K. pneumoniae* strains [27], does not correspond with the expression results in this study using an in vitro infection model in A549 and A431 cell lines with the same strains. Interestingly, *cnf-1* was not expressed in any of the strains, whereas *magA* was expressed in only one strain. This observed inconsistency in detection and expression is probably attributable to the cell lines used in our study since prior reports have demonstrated differential levels of expression according to the infected cell type. One example is that *cnf-1* tends to have a higher frequency of expression during urinary tract infections [41], whereas *magA* expression has been linked to hypermucoviscosity in strains associated with liver abscesses [42]. Furthermore, the temporal dynamics of the infectious process have a profound effect on gene expression. Some genes are known to be expressed early in the infection, while others have later expression. For instance, the early-expressed type 1 fimbrial genes are regulated by an invertible element located in the promoter region [43]. This element permits gene expression to be repressed or activated based on the prevailing conditions of the infection process.

PFGE is a frequently used method for clonal typing of isolates in outbreaks and allows for source identification [44]. In this study, 13 different PFtypes distributed in Subtypes B1 and B2 of Clade B were identified, some of them integrated by two, three, four, and up to eight strains. In Clade A, 19 PFtypes distributed in Subtypes A1 and A2 were identified. These comprehensive findings showed different virulome expression patterns associated with the antibiotic-resistance genotype and clinical origin in *K. pneumoniae* strains from the same PFtype, where the largest PFtype (subtype A2) was found to comprise nine strains (12, 61, 73, 76, 94, 101, 112, 124, and 133) isolated from hospital-acquired (bacteremia [n = 3]) and community-acquired infections (respiratory infection [n = 3], and infected ulcers [n = 2] and UTI [n = 1]), all with different virulence gene expression profiles associated with the antibiotic resistance genotype.

The results showed that *K. pneumoniae* strains from the same PFtype, isolated from different patients with hospital or community infections, are possibly found in the hospital environment. Therefore, some patients with community-acquired strains could have acquired these strains during their hospital stay.

The unsupervised hierarchical clustering analysis, similar to the PFGE method, revealed a diverse spectrum of virulome expression patterns associated with the genotype-phenotype characteristics of antibiotic resistance and the origin of *K. pneumoniae* strains isolated from patients with either community- or hospital-acquired infections. High throughput technologies, such as whole genome sequencing, could provide a more comprehensive understanding of the global composition of the virulence and antimicrobial-resistance genes. These analyses will be performed in future works. To the best of our knowledge, this is the first study performed in Mexico where the wide distribution of different virulome expression patterns associated with the antibiotic resistance genotype in *K. pneumoniae* strains isolated from patients with different hospital- and community-acquired infections is shown, suggesting the high pathogenic potential of these strains during infections.

## 4. Materials and Methods

### 4.1. Origin of the Strains

A total of 150 strains of *K. pneumoniae* were analyzed, which were collected from September 2019 to March 2020 in the Microbiology laboratory of the Hospital General Regional No. 72 (Instituto Mexicano del Seguro Social, State of Mexico, Mexico) located in the municipality of Tlalnepantla de Baz, Edo. from Mexico, Mexico. The strains were isolated from samples of hospitalized patients with ongoing infections, such as bacteremia (n = 21) and pneumonia (n = 4), which they acquired during their hospital stay after admission for managing complications arising from other comorbidities such as diabetes, high blood pressure, chronic obstructive pulmonary disease, and obesity. Additionally, strains were collected from the non-hospitalized external community, encompassing UTIs (urinary tract infection; n = 61), respiratory (n = 53), ulcers (n = 8) and other infections (stool culture (n = 2), and tumor biopsy (n = 1)). All patients provided informed consent for inclusion in this study. The study was approved by the institution’s Ethics Committee (identification code: R-2022-1406-033).

### 4.2. Bacterial DNA Extraction

The DNA of the *K. pneumoniae* strains was extracted by the boiling method described previously [45]. Each of the strains was seeded on eosin methylene blue agar (EMB; DIBICO, Edo. de México, Mexico) at 37 °C for 12 h under constant agitation. Four isolated 2-mm (in diameter) colonies were collected, suspended in Eppendorf tubes with 200 µL of sterile water, incubated at 100 °C for 10 min, and centrifuged at 10,000× *g* for 5 min. DNA obtained in the supernatant was stored at −20 °C. In order to use 100 ng/µL DNA for the PCR reaction, DNA concentration and purity were measured using a NanoDrop 2000 spectrophotometer (Thermo Fisher Scientific, Waltham, MA, USA).

### 4.3. Identification of K. pneumoniae through Polymerase Chain Reaction (PCR)

*K. pneumoniae* strains were identified using PCR amplification of 16S–23S rDNA internal transcribed spacer [46]. The final volume per reaction mixture for each uniplex PCR assay was 20 µL; 1 µL of forward primer and 1 µL of reverse primer (10 pmol, Integrated DNA Technologies, San Diego, CA, USA), 12 µL of Taq DNA Polymerase 2× Master Mix RED (AMPLIQON, Copenhagen, Denmark), 3 µL of nuclease-free water, and 3 µL of DNA template (100 ng). The PCR amplification conditions were performed using a T100™ Thermal Cycler (Bio-Rad, Feldkirchen, Germany) instrument as follows; initial denaturation at 94 °C for 5 min, 30 cycles of denaturation at 94 °C for 30 s, annealing at 55 °C for 30 s, and extension at 72 °C for 50 s; with a final extension of 72 °C for 5 min. The *K. pneumoniae* ATCC 700721 strain was used as a positive control. The amplified DNA fragments (130 pb) were visualized using 2% agarose gels stained with Midori Green Direct (Nippon Genetics, Düren, Germany).

### 4.4. Multidrug-Resistance Phenotype

The Kirby–Bauer disk diffusion method (Diagnostic Research, CDMX, Mexico) was used to determine resistance to the following 12 antimicrobials; carbenicillin (CB; 100 µg), ampicillin (AM; 10 µg), cefotaxime (CFX; 30 µg), cephalothin (CF; 30 µg), gentamicin (GE; 10 µg), amikacin (AK; 30 µg), netilmicin (NET; 30 µg), ciprofloxacin (CPF; 5 µg), norfloxacin (NOF; 10 µg), nitrofurantoin (NF; 300 µg), chloramphenicol (CL; 30 µg), and trimethoprim with sulfamethoxazole (SXT; 25 µg). For the reproducibility of the method, the *Escherichia coli* ATCC 25922 strain was used as a control. The criteria established by the Clinical and Laboratory Standards Institute were used for the interpretation of the results [47]. *K. pneumoniae* strains were classified as multidrug resistant (MDR) when they presented resistance to more than three antibiotics from different groups.

### 4.5. Detection of Antibiotic-Resistance Genes Using Conventional PCR

The conventional polymerase chain reaction (PCR) method was used to determine on the chromosome of the *K. pneumoniae* strains; the presence or absence of the genes encode resistance to beta–lactams (*bla_SHV_*, *bla_CITM_*, *bla_TEM_*, *bla_CTX-1_*, *bla_CTXM-2_*, and *bla_CTXM-9_*), streptomycin (*aadA1*), gentamicin (*aac(3)-IV*), sulfonamide (*sul1*), chloramphenicol (*cat1* and *cmlA*), tetracycline (*tet(A)* and *tet(B)*), trimethoprim (*dfrA1*), and quinolones (*qnr*). The primers and PCR conditions used for detection of antibiotic resistance genes were as described previously [48,49]. The final volume-per-reaction mixture for each uniplex PCR assay was 20 µL; 1 µL of forward primer and 1 µL of reverse primer (10 pmol, Integrated DNA Technologies, San Diego, CA, USA), 12 µL of Taq DNA Polymerase 2× Master Mix RED (AMPLIQON, Copenhagen, Denmark), 3 µL of nuclease- free water, and 3 µL of DNA template (100 ng). The *K. pneumoniae* strains from clinical isolates Kp7, Kp17, Kp37, Kp39, and Kp74, which harbor the antibiotic-resistance genes studied, were used as positive controls.

### 4.6. Bacterial Dilution of Strains for Infection of Human Epithelial Cell Lines

One colony (2 mm in diameter) of every *K. pneumoniae* strain was seeded separately in brain heart infusion broth (MCD Lab, Edo. de México, Mexico) and incubated for 12 h at 37 °C under constant shaking. Following the instructions in the RNA protect Bacteria Reagent Manual (Qiagen, Hilden, Germany), a 1:4 dilution of each *K. pneumoniae* culture was performed using phosphate-buffered saline solution to obtain an optical density at 600 nm = 1.0 using a Beckman DU-7400 spectrophotometer (Laguna Hills, CA, USA), which corresponded to a concentration of 1 × 10^9^ cells/mL. Subsequently, dilutions were performed in order to obtain a concentration of 2 × 10^6^ cells/mL.

### 4.7. Preparation of Human Epithelial Cell Lines and In Vitro Infection Model

The in vitro infection model using human epithelial cell lines to determine virulence gene expression was performed as previously described [50,51]. With the aim of emulating a human infection process, we prepared cultures of human lung cancer A549 (ATCC CCL-185) and human epidermoid A431 (CRL-1555, Manassas, VA, USA) cell lines. The cell lines were plated in DMEM medium (Dulbecco’s Modified Eagle Medium, Corning NY, USA) and supplemented with 10% FBS (Fetal Bovine Serum, Corning NY, USA) in a 24-well cell culture plate, until a confluency of 1.8 × 10^5^ cells/well was reached for the in vitro infection experiment. All cultures were incubated at 37 °C under 5% of CO_2_ atmosphere.

To stimulate the bacterial infection of the human cell lines, each dilution of *K. pneumoniae* culture isolated from patients with bacteremia, pneumonia, and respiratory, ulcer, and other infections were inoculated (50 µL; 2 × 10^6^ cells/mL) onto the surface of a monolayer of the cultured 1.8 × 10^5^ cells of the epithelial cell line A549 (ATCC CCL-185) derived from lung cancer, while dilutions (50 µL; 2 × 10^6^ cells/mL) of *K. pneumoniae* from patients with UTIs were inoculated on the monolayer of the human epidermoid cell line A431 (CRL-1555, Manassas, VA, USA). The 24-well plates were incubated with 1 mL of F12K plus 10% fetal bovine serum at 37 °C for 48 h under a 5% CO_2_ atmosphere with saturated humidity. The maintenance medium (F12K plus 10% fetal bovine serum) was changed every 24 h. The *K. pneumoniae* strain, ATCC 700721, was used as a control.

### 4.8. K. pneumoniae RNA Extraction and Reverse-Transcription to cDNA

To establish the expression of virulence genes, RNA was extracted from *K. pneumoniae* strains after in vitro infection of human epithelial cell lines, for which *K. pneumoniae* strains were harvested from the surface of A549 and A431 cell line cultures and suspended in 1000 µL of RNA Protect Bacteria reagent (Qiagen). Samples were centrifuged at 8000× *g* for 10 min to obtain bacterial cell sediment. RNA was extracted from the bacterial sediment using a QIAcube automated extraction equipment (Qiagen, Hilden, Germany) with the commercial RNeasy Mini Kit (Qiagen), which involved bacterial lysis with TE buffer (10 mM Tris-HCl, 1 mM EDTA, pH 8) containing 1 mg/mL lysozyme. A NanoDrop 2000 spectrophotometer (Thermo Fisher Scientific, Waltham, MA, USA, EE. UU) was used to determine the concentration and purity of total RNA. The Reverse Transcription QuantiTect kit (Qiagen) was used to perform first-strand cDNA synthesis.

### 4.9. Determination of K. pneumoniae Virulome Expression Using Real-Time PCR

To determine virulome expression in *K. pneumoniae* strains by real-time PCR, the Rotor-Gene Q 5plex HRM device (Qiagen, Hilden, Germany) was used. The primers used to assess the expression of the virulence genes of *K. pneumoniae virulence genes* encoding for adhesins, *fimH-1* (type 1 fimbriae), *mrkD* (type 3 fimbriae), *kpn* (fimH-type adhesin), and *ycfM* (outer membrane lipoprotein); iron-acquisition systems, *entB* (enterobactin biosynthesis), *irp1*, *irp2*, *ybtS* (yersiniabactin biosynthesis), *fyuA* (yersiniabactin receptor), *iutA* (aerobactin receptor), and *iroN* (catecholate siderophore receptor); protectins, *magA* (mucoviscosity-associated gene A) and *rmpA* (mucoid regulator phenotype A), and toxins; *hlyA* (hemolysin) and *cnf-1* (necrotizing cytotoxic factor 1) were as previously described [15]. The final volume-per-reaction mixture for each real-time PCR assay was 20 µL: 12.5 µL of the Rotor–Gene SYBR Green PCR kit master mix (Qiagen), 1 µL of each forward and reverse primer (1 µM), 8.5 µL of RNAase-free water, and 2 µL of the cDNA (100 ng). Amplification was performed at 95 °C for 5 min, followed by 40 cycles of 95 °C for 5 s, and an extension at 60 °C for 10 s. For each quantitative real-time PCR (qPCR) assay, we performed a standard curve prepared from three dilutions of cDNA (100, 200, and 300 ng/µL) from a strain harboring the gene of interest (positive control). From these three cDNA dilutions, the Rotor–Gene Q 5plex HRM System software version 2.3.1.49 (Qiagen) calculated the cutoff for the definition of the threshold cycle (TC) of each strain. In this way, with the CT values of each strain obtained with respect to the CT of the standard curve of the control strain, the arithmetic mean was obtained, and the expression percentage of each gene was determined. Each real-time PCR assay included a melting curve, a housekeeping gene (*rpoB*), and a non-templated control (NTC). The *K. pneumoniae strains* from clinical isolates Kp1, Kp6, and Kp12, harboring the 15 virulence genes studied (*fimH-1*, *mrkD*, *kpn, ycfM, entB*, *irp1*, *irp2*, *ybtS*, *fyuA*, *iutA*, *iroN, magA rmpA*, *hlyA*, and *cnf-1*) were used as positive controls for the preparation of the standard curve.

### 4.10. Pulse Field Gel Electrophoresis (PFGE)

The preparation of agarose blocks with *K. pneumoniae* genomic DNA was performed using the method previously described by the standardized PFGE protocol PulseNet, USA [52]. To separate *XbaI* fragments, electrophoresis was performed on 1% of agarose gels and 0.5X TBE buffer at 14 °C with an increased pulse time from 2.2 to 54.2 s for 20 h and 6.0 V/cm using the CHEF–MAPPER (Bio-Rad, Hercules, CA, USA) device. *XbaI* fragment sizes were estimated using *XbaI* fragments of *Salmonella enterica* serotype Braenderup global standard H9812. The Gel Logic 112 image photo documentation system (Kodak, Rochester, NY, USA) was used to digitize the images. The BioNumerics v.7.1 software package (Applied Maths, Sint Martens-Latem, Belgium) was used to analyze the fingerprint profile on the PFGE gel. After background subtraction and gel normalization, the fingerprints in the profiles were typed based on band similarity and dissimilarity, using Dice’s similarity coefficient and the unweighted pairwise clustering method with arithmetic mean according to average linkage clustering methods. PFGE patterns were interpreted according to the criteria previously described for the typing of bacterial strains [53]. (1) Identical: Strains are designated as genetically indistinguishable if their restriction patterns have the same number of bands. (2) Closely related: The PFGE patterns of the strains present two or three bands of difference. (3) Possibly related: The PFGE pattern of the strains differs between four and six bands. (4) Unrelated: The PFGE pattern differs in seven or more bands.

### 4.11. Unsupervised Hierarchical Clustering

*K. pneumoniae* strains were systematically grouped according to virulome expression, antibiotic-resistance genotype, antibiotic-resistance phenotype, diagnosis, and clinical origin from patients with hospital- and community-acquired infections using unsupervised hierarchical clustering based on Euclidean distances for categorical variables. A categorical data matrix that included virulence gene expression, antibiotic-resistance genotype, antibiotic-resistance phenotype (resistance and sensible), and the clinical origin of patients with hospital-acquired infections and community-acquired infections, as well as diagnosis, was created in R (v.3.6.2). The distance of each strain was calculated based on the overall similarity coefficient, which estimates the maximum possible absolute discrepancy between each combined pair of strains. Strains were visualized in a distribution diagram with a dendrogram constructed using ComplexHeatmap (v3.6.2, R core).

### 4.12. Statistical Analysis

The χ^2^ test was applied using the SPSS statistical software (version 20.0; SPSS Inc., Chicago, IL, USA) (*p* < 0.05) to establish differences between the frequency of expression of virulence genes related to the antibiotic-resistance genotype and clinical origin in *K. pneumoniae* strains isolated from patients with hospital- and community-acquired infections.

## 5. Conclusions

Our results highlighted the existence of different PFtypes of *K. pneumoniae* causing hospital- and community-acquired infections, with wide virulome expression profiles related to the antimicrobial-resistance genotype and infection type. Therefore, it is important to implement new monitoring and control programs on the emergence of new hypervirulent and MDR *K. pneumoniae* strains to improve medical treatment and reduce mortality rates.

## Figures and Tables

**Figure 1 ijms-24-14754-f001:**
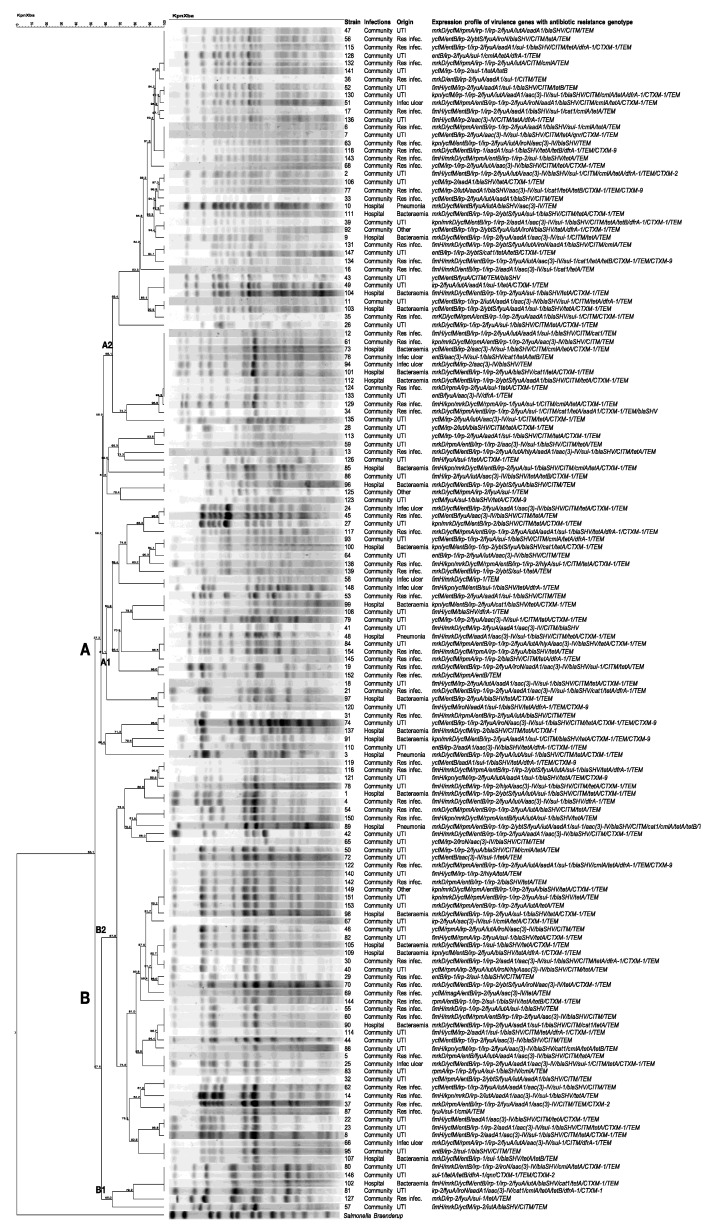
Pulse-field gel electrophoresis (PFGE) profile dendrogram and expression patterns of virulence genes and antibiotic-resistance genotyping in *K. pneumoniae* strains from hospital- and community-acquired infections. The dendrogram was produced through the Dice similarity coefficient and unweighted pairwise clustering method with arithmetic mean clustering methods using PFGE imaging of *XbaI-digested* genomic DNA. The scale bar shows the correlation coefficient (%). Capital letters (**A**,**B**) represent the two main clades produced. The letters B1, B2, A1, and A2 represent subgroups of strains. The strain number, type of infection, origin, and virulence gene expression pattern associated with the antibiotic resistance genotype are shown on the right.

**Figure 2 ijms-24-14754-f002:**
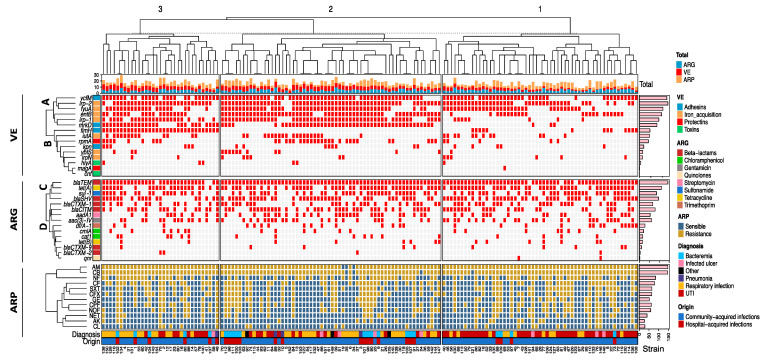
Hierarchical grouping of *K. pneumoniae* strains according to virulome expression related to antibiotic-resistance genotype–phenotype, diagnosis, and clinical origins. Positivity and negativity for a particular genotype is represented by a red and grey rectangle, respectively. Upper panel: Virulome expression (VE). Middle panel: Antibiotic-resistance genotype (ARG). Lower panel: Antibiotic-resistance phenotype (ARP). Left axis: Expression of virulence factors (Clades A and B) and detected antibiotic-resistance genes (Clades D and C). Right axis: Absolute frequency of expression by gene and antibiotic resistance. Top: Cladogram total of VE, ARG, and ARP of the strains. Bottom: Diagnosis and origin of the strains. ARP section: AM = ampicillin, CB = carbenicillin, NF = nitrofurantoin, CF = cephalothin, SXT = trimethoprim-sulfamethoxazole, CFX = cefotaxime, GE = gentamicin, CPF = ciprofloxacin, NOF = norfloxacin, NET = netilmicin, AK = amikacin and CL = chloramphenicol.

**Table 1 ijms-24-14754-t001:** Antibiotic resistance phenotype in strains of *Klebsiella pneumoniae* isolated from patients with hospital- and community-acquired infections.

Antibiotics	Antibiotic Group	Number of Resistant Strains (%)
Carbenicillin (CB)	beta-lactams	148 (98.6)
Ampicillin (AM)		148 (98.6)
Cefotaxime (CFX)		66 (44)
Cephalothin (CF)		81 (54)
Gentamicin (GE)	Aminoglycosides	63 (42)
Amikacin (AK)		28 (18.6)
Netilmycin (NET)		42 (28)
Ciprofloxacin (CPF)	Quinolones	64 (42.6)
Norfloxacin (NOF)		61 (40.6)
Nitrofurantoin (NF)	Nitrofurans	86 (57.3)
Chloramphenicol (CL)	Phenicols	31 (20.6)
Sulfamethoxazole/trimethoprim (SXT)	Sulfonamide/Trimethoprim	67 (44.6)

**Table 2 ijms-24-14754-t002:** Frequency of expression of virulence genes in strains of *Klebsiella pneumoniae* isolated from patients with hospital- and community-acquired infections.

Origin of the Strains		Number of Strains (%)														
		Adhesins				Iron Acquisition Systems							Protectins		Toxins	
		*fimH*	*mrkD*	*kpn*	*ycfM*	*entB*	*irp1*	*irp2*	*ybtS*	*fyuA*	*iutA*	*iroN*	*magA*	*rmpA*	*hlyA*	*cnf-1*
Hospital-acquired (n = 25)	Bacteremia (n = 21)	5 (23.8)	15 (71.4)	5 (23.8)	21 (100)	19 (90.5)	14 (66.7)	19 (90.5)	6 (28.6)	17 (81)	2 (9.5)	0	0	0	0	0
	Pneumonia (n = 4)	1 (25)	4 (100)	0	4 (100)	3 (75)	2 (50)	2 (50)	1 (25)	3 (75)	3 (75)	0	0	1 (25)	0	0
Community-acquired (n = 125)	UTI (n = 61)	22 (36.1)	13 (21.3)	6 (9.8)	45 (73.8)	27 (44.3)	27 (44.3)	52 (85.2)	2 (3.3)	38 (62.3)	17 (27.9)	7 (11.5)	0	11 (18)	4 (6.6)	0
	Respiratory infection (n = 53)	16 (30.2)	37 (69.8)	6 (11.3)	40 (75.5)	41 (77.4)	36 (67.9)	45 (84.9)	5 (9.4)	38 (71.7)	20 (37.7)	5 (9.4)	1 (1.9)	23 (43.4)	2 (3.8)	0
	Infected ulcer (n = 8)	2 (25)	6 (75)	1 (12.5)	7 (87.5)	5 (62.5)	4 (50)	5 (62.5)	0	4 (50)	1 (12.5)	1 (12.5)	0	2 (25)	0	0
	* Other (n = 3)	0	2 (66.7)	1 (33.3)	3 (100)	2 (66.7)	2 (66.7)	3 (100)	1 (33.3)	3 (100)	1 (33.3)	1 (33.3)	0	2 (66.7)	0	0
*p*-value		0.732	**1.8 × 10^−8^**	0.264	0.054	**0.0001**	0.103	0.422	**0.009**	0.265	0.058	0.210	0.593	**0.0001**	0.763	-
Total (n = 150)		46 (30.7)	77 (51.3)	19 (12.7)	120 (80)	97 (64.7)	85 (56.7)	126 (84)	15 (10)	103 (68.7)	44 (29.3)	14 (9.3)	1 (0.7)	39 (26)	6 (4)	0

Significant *p*-values (<0.05) are shown in bold. * Other: stool culture (n = 2) and tumor biopsy (n = 1).

**Table 3 ijms-24-14754-t003:** Frequencies of antibiotic-resistance genes in strains of *Klebsiella pneumoniae* isolated from patients with hospital- and community-acquired infections.

Origin of the Strains		Number of Strains (%)														
		Beta–Lactams						Streptomycin	Gentamicin	Sulfonamide	Chloramphenicol		Tetracycline		Trimethoprim	Quinolones
		*bla_SHV_*	*bla_CITM_*	*bla_TEM_*	*bla_CTXM-1_*	*bla_CTXM-2_*	*bla_CTXM-9_*	*aadA1*	*aac(3)-IV*	*sul1*	*cat1*	*cmlA*	*tet(A)*	*tet(B)*	*dfrA1*	*qnr*
Hospital-acquired (n = 25)	Bacteremia (n = 21)	16 (76.2)	10 (47.6)	20 (95.2)	17 (80.9)	0	1 (4.7)	5 (23.8)	2 (9.5)	12 (57.1)	5 (23.8)	2 (9.52)	20 (95.2)	1 (4.7)	1 (4.7)	0
	Pneumonia (n = 4)	2 (50)	3 (75)	4 (100)	2 (50)	0	0	2 (50)	3 (75)	3 (75)	1 (25)	1 (25)	3 (75)	1 (25)	0	0
Community-acquired (n = 125)	UTI (n = 61)	39 (63.9)	36 (59)	56 (91.8)	30 (49)	3 (4.9)	4 (6.5)	20 (32.7)	32 (52.4)	32 (52.4)	3 (4.9)	10 (16.3)	46 (75.4)	8 (13.1)	14 (22.9)	2 (3.2)
	Respiratory infection (n = 53)	32 (60.3)	26 (49)	53 (100)	13 (24.5)	1 (1.8)	5 (9.4)	24 (45.2)	19 (35.8)	34 (64.1)	7 (13.2)	7 (13.2)	35 (66)	4 (7.5)	10 (18.8)	0
	Infected ulcer (n = 8)	5 (62.5)	4 (50)	8 (100)	3 (37.5)	0	0	3 (37.5)	5 (62.5)	4 (50)	1 (12.5)	1 (12.5)	5 (62.5)	1 (12.5)	2 (25)	0
	Other (n = 3)	2 (66.6)	0	3 (100)	2 (66.6)	0	0	0	0	1 (33.3)	0	0	2 (66.6)	0	1 (33.3)	0
*p*-value		0.840	0.389	0.363	**0.0001**	0.765	0.941	0.363	**0.001**	0.709	0.125	0.906	0.094	0.565	0.334	0.710
Total (n = 150)		96 (64)	79 (52.6)	144 (96)	67 (44.7)	4 (2.7)	10 (6.7)	54 (36)	61 (40.7)	86 (57.3)	17 (11.3)	21 (14)	111 (74)	15 (10)	28 (18.7)	2 (1.3)

Significant *p*-values (<0.05) are shown in bold.

**Table 4 ijms-24-14754-t004:** Distribution of virulence genes expression according to antibiotic resistance genes in strains of *Klebsiella pneumoniae* isolated from patients with hospital-acquired infections.

Hospital-Acquired Infections (n = 25)																
Antibiotic Resistance Gene		Number of Strains (%)														
		Adhesins				Iron Acquisition Systems							Protectins		Toxins	
		*fimH*	*mrkD*	*kpn*	*ycfM*	*entB*	*irp1*	*irp2*	*ybtS*	*fyuA*	*iutA*	*iroN*	*magA*	*rmpA*	*hlyA*	*cnf-1*
Beta–lactams	*bla_SHV_*	6 (24)	18 (72)	4 (16)	24 (96)	18 (72)	15 (60)	20 (80)	7 (28)	18 (72)	5 (20)	0	0	1 (4)	0	0
	*bla_CITM_*	4 (16)	12 (48)	2 (8)	13 (52)	10 (40)	8 (32)	12 (48)	5 (20)	10 (20)	3 (12)	0	0	1 (4)	0	0
	*bla_TEM_*	5 (20)	22 (88)	5 (20)	24 (96)	22 (88)	16 (64)	20 (80)	7 (28)	20 (80)	5 (20)	0	0	1 (4)	0	0
	*bla_CTXM-1_*	6 (24)	13 (52)	5 (20)	19 (76)	16 (64)	11 (44)	17 (68)	5 (20)	15 (60)	3 (12)	0	0	0	0	0
	*bla_CTXM-2_*	0	0	0	0	0	0	0	0	0	0	0	0	0	0	0
	*bla_CTXM-9_*	0	1 (4)	1 (4)	0	1 (4)	0	1 (4)	0	1 (4)	0	0	0	0	0	0
Streptomycin	*aadA1*	1 (4)	6 (24)	1 (4)	7 (28)	6 (24)	5 (20)	6 (24)	3 (12)	6 (24)	1 (4)	0	0	1 (4)	0	0
Gentamicin	*aac(3)-IV*	1 (4)	4 (16)	0	5 (20)	4 (16)	2 (8)	3 (12)	1 (4)	3 (12)	2 (8)	0	0	1 (4)	0	0
Sulfonamide	*sul1*	4 (16)	13 (52)	2 (8)	15 (60)	13 (52)	11 (44)	12 (48)	3 (12)	11 (44)	3 (12)	0	0	1 (4)	0	0
Chloramphenicol	*cat1*	1 (4)	4 (16)	1 (4)	5 (20)	5 (20)	5 (20)	5 (20)	2 (8)	5 (20)	2 (8)	0	0	1 (4)	0	0
	*cmlA*	1 (4)	2 (8)	1 (4)	3 (12)	3 (12)	1 (4)	3 (12)	1 (4)	2 (8)	1 (4)	0	0	1 (4)	0	0
Tetracycline	*tet(A)*	6 (24)	14 (56)	5 (20)	21 (84)	15 (60)	14 (56)	20 (80)	5 (20)	18 (72)	4 (16)	0	0	3 (12)	0	0
	*tet(B)*	0	2 (8)	0	2 (8)	2 (8)	1 (4)	1 (4)	1 (4)	1 (4)	0	0	0	1 (4)	0	0
Trimethoprim	*dfrA1*	0	0	1 (4)	1 (4)	1 (4)	0	1 (4)	0	1 (4)	0	0	0	0	0	0
Quinolones	*qnr*	0	0	0	0	0	0	0	0	0	0	0	0	0	0	0
*p*-value		**7.1 × 10^−11^**	**0.0004**	**1.2 × 10^−7^**	**0.0004**	**0.0004**	**0.0004**	**0.0004**	**0.0004**	**0.0004**	**0.0004**	-	-	1	-	-

Significant *p*-values (<0.05) are shown in bold.

**Table 5 ijms-24-14754-t005:** Distribution of virulence genes expression according to antibiotic resistance genes in strains of *Klebsiella pneumoniae* isolated from patients with community-acquired infections.

Community-Acquired Infections (n = 125)																
Antibiotic Resistance Gene		Number of Strains (%)														
		Adhesins				Iron-Acquisition Systems							Protectins		Toxins	
		*fimH*	*mrkD*	*kpn*	*ycfM*	*entB*	*irp1*	*irp2*	*ybtS*	*fyuA*	*iutA*	*iroN*	*magA*	*rmpA*	*hlyA*	*cnf-1*
Beta-lactams	*bla_SHV_*	31 (24.8)	40 (32)	8 (6.4)	72 (57.6)	56 (44.8)	51 (40.8)	76 (60.8)	5 (4)	58 (46.4)	28 (22.4)	12 (9.6)	0	26 (20.8)	3 (2.4)	0
	*bla_CITM_*	18 (14.4)	30 (24)	6 (4.8)	56 (44.8)	41 (32.8)	38 (30.4)	59 (47.2)	3 (2.4)	45 (36)	23 (18.4)	8 (6.4)	0	18 (14.4)	3 (2.4)	0
	*bla_TEM_*	38 (30.4)	57 (45.6)	14 (11.2)	91 (72.8)	73 (58.4)	65 (52)	98 (78.4)	8 (6.4)	75 (60)	39 (31.2)	12 (9.6)	1 (0.8)	35 (28)	5 (4)	0
	*bla_CTXM-1_*	14 (11.2)	20 (16)	6 (4.8)	36 (28.8)	26 (20.8)	27 (21.6)	42 (33.6)	3 (2.4)	30 (24)	12 (9.6)	6 (4.8)	0	9 (7.2)	2 (1.6)	0
	*bla_CTXM-2_*	1 (0.8)	1 (0.8)	0	1 (0.8)	2 (1.6)	2 (1.6)	2 (1.6)	0	2 (1.6)	1 (0.8)	0	0	1 (0.8)	0	0
	*bla_CTXM-9_*	3 (2.4)	3 (2.4)	1 (0.8)	10 (8)	5 (4)	4 (3.2)	5 (4)	0	5 (4)	4 (3.2)	2 (1.6)	0	1 (0.8)	0	0
Streptomycin	*aadA1*	15 (12)	23 (18.4)	4 (3.2)	39 (31.2)	31 (24.8)	26 (20.8)	40 (32)	2 (1.6)	32 (25.6)	17 (13.6)	5 (4)	0	10 (8)	1 (0.8)	0
Gentamicin	*aac(3)-IV*	17 (13.6)	23 (18.4)	6 (4.8)	41 (32.8)	35 (28)	29 (23.2)	48 (38.4)	1 (0.8)	36 (28.8)	18 (14.4)	8 (6.4)	1 (0.8)	8 (6.4)	3 (2.4)	0
Sulfonamide	*sul1*	24 (19.2)	32 (25.6)	9 (7.2)	52 (41.6)	42 (33.6)	41 (32.8)	58 (46.4)	2 (1.6)	46 (36.8)	18 (14.4)	3 (2.4)	0	15 (12)	3 (2.4)	0
Chloramphenicol	*cat1*	4 (3.2)	4 (3.2)	1 (0.8)	6 (4.8)	6 (4.8)	6 (4.8)	8 (6.4)	1 (0.8)	6 (4.8)	1 (0.8)	1 (0.8)	0	1 (0.8)	0	0
	*cmlA*	6 (4.8)	7 (5.6)	3 (2.4)	12 (9.6)	8 (6.4)	14 (11.2)	16 (12.8)	1 (0.8)	17 (13.6)	5 (4)	4 (3.2)	0	4 (3.2)	0	0
Tetracycline	*tet(A)*	26 (20.8)	35 (28)	12 (9.6)	64 (51.2)	48 (38.4)	48 (38.4)	71 (56.8)	5 (4)	52 (41.6)	24 (19.2)	9 (7.2)	1 (0.8)	21 (16.8)	5	0
	*tet(B)*	4 (3.2)	2 (1.6)	2 (1.6)	7 (5.6)	4 (3.2)	7 (5.6)	10 (8)	1 (0.8)	5 (4)	3 (2.4)	0	0	0	0	0
Trimethoprim	*dfrA1*	7 (5.6)	10 (8)	1 (0.8)	20 (16)	18 (14.4)	15 (12)	19 (15.2)	2 (1.6)	14 (11.2)	9 (7.2)	3 (2.4)	0	5 (4)	1 (0.8)	0
Quinolones	*qnr*	0	0	0	1 (0.8)	1 (0.8)	0	1 (0.8)	0	1 (0.8)	0	0	0	0	0	0
*p*-value		**0.0004**	**0.0004**	**0.0004**	**0.0004**	**0.0004**	**0.0004**	**0.0004**	**0.0004**	**0.0004**	**0.0004**	**0.0004**	1	**0.0004**	**0.0004**	-

Significant *p*-values (<0.05) are shown in bold.

## Data Availability

Not applicable.

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
