# Peer review of "Extensive Expression of the Virulome Related to Antibiotic Genotyping in Nosocomial Strains of Klebsiella pneumoniae"

_ijms, 2023, doi:10.3390/ijms241914754_

Round 1

Reviewer 1 Report

In the manuscript the Authors present their findings concerning the virulence related genes of K. pneumoniae in the isolate collection of clinical origin from Mexico. The data collected is important to the scientific community, however, i have some concerns with presentation, which makes the understanding of the experiments performed difficult. My major concerns are listed below.

Major remarks:

1. The abstract does not clearly present the article. A few sentences more should be written in the introduction, clarifying not only the problem, but also the method - it is unclear, how expression was measured, since it is only presented as percentage of isolates (also see remarks below).

2. The writing of the beta lactamase names is inconsistent, and should be corrected throughout the text. 

3. One of the keywords is "multidrug resistant", however, there is no information about phenotypical resistance in the presented data.

4. The introduction is too short, and does not clearly present the Authors' motivation to perform research. Also, the virulence genes are presented, but no explanation of their choice or importance given. It must be given in more general terms in Introduction, and could be even more detailed in Results or Discussion.

5. Why are siderophores described together with capsular virulence factors, instead of grouping them with iron acquisition factors?

6. Overall, methods used, and data presented is inconsistent - Introduction presents in vitro model used, and Methods section also describes it, but there is no mentioning it at all in the Results section. It should be clear, which Methods were used to get each result.

7. In all the cases, if the Authors claim the data was significant or insignificant, the value given in the text is always "p<0.05". E.g. lines 72, 84...

8. As mentioned before, the Methods used to get Results are unclear. The Authors present very briefly that in vitro tests were done, also regular PCR, and q(rt)PCR was performed, but how the percentages of isolated expressing virulence genes were calculated remains unclear. This must be explained.

9. It is unclear, how the feature "hospital acquired" was attributed to the isolates, the Authors should explain it.

10. What were the principles of separating the pulsotypes into separate clades and groups, described in the text? I. e. why did the Authors chose to describe the groups they selected, but not the bigger or smaller branches?

11. Also, all the pulsotype groups consist of a mix of hospital and community acquired isolates. This raises a question, already mentioned in the remarks, were the hospital acquired infections really nosocomial? This should be discussed in the text.

12. Lines 154-157 discusses information that was not presented in the Results section.

13. In lines 217-219 the Authors present mortality data from a reference, however, it is unclear, what information do the Authors have about the destinies of the patients, from whom the K. pneumoniae used in this study was collected.

14. Was the amplification efficiency of the primers used for qPCR measured?

15. It is unclear, how the expression data was calculated - what calculations were done after Ct detection?

16. In the Conclusions and in the text, the Authors present that "clones" of K. pneumoniae could be detected. Is the presented PFGE data enough to claim the presence of clones? Also, as long as it is unclear how hospital-acquired infections were attributed to being hospital-acquired, the conclusions cannot be considered.

Minor remarks:

1. In tables, the "Virulence gene expression no. (%)" and similar column titles "no." is not the correct term.

2. The title of Table 2 is of wrong formatting and fused to the main text.

3. Pulsotypes term abbreviation is explained several times in the text, and abbreviated differently. It should be made uniform. 

4. The information in Fig. 1 is barely readable - the genes found in the isolates could be presented clearer using Heatmap or similar visualization.

5. The hemolysin gene name is incorrect in lines 180, 185

6. In the Methods lines 256-257 the Authors present, that DNA content of crudely lysed cells was measured by NanoDrop. What is the purpose and reason of that...?

7. Line 256 should be -20 degrees, not ~20..?

The English language is fine; some phrases could be made clearer, but they do not interfere with understanding of the text.

Author Response

ID de manuscrito: ijms-2553055

Extensive expression of the virulome related to antibiotic genotyping
in nosocomial strains of Klebsiella pneumoniae

Response letter
We are thankful to the reviewers for this opportunity to improve our
manuscript. We appreciate the insightful and constructive comments, and the
time spent on their review. In this revised version we addressed all review
comments and have modified the text accordingly. These additions,
clarifications and modifications have improved significantly the quality of the
manuscript. The following reply addresses these important issues point by
point. Our answers are written in bold script.

Reviewer comments:
Reviewer 1

Major remarks:
1. The abstract does not clearly present the article. A few sentences more
should be written in the introduction, clarifying not only the problem, but also
the method - it is unclear, how expression was measured, since it is only
presented as percentage of isolates (also see remarks below).

Thank you for the observation and for this valuable comment. The
abstract was restructured, expanded and improved. The abstract now
clearly presents the research article.

2. The writing of the beta lactamase names is inconsistent, and should be
corrected throughout the text.

We apologize for this inconsistency. We have uniformly corrected the
name of the beta-lactamase gene names throughout the text.

3. One of the keywords is "multidrug resistant", however, there is no
information about phenotypical resistance in the presented data.

Thank you for the observation. We have included in the results an
additional table with information on the phenotype of resistance to
antibiotics in the strains (table 1), as well as the percentage of
multidrug-resistant (MDR) strains. Consequently, this information has
been included in the abstract and has also been analyzed in the
discussion.

4. The introduction is too short, and does not clearly present the Authors'
motivation to perform research. Also, the virulence genes are presented,
but no explanation of their choice or importance given. It must be given in
more general terms in Introduction, and could be even more detailed in
Results or Discussion.

We appreciate your observation and kind comment. We have expanded
the introduction, now describing with greater clarity and detail the
importance of the different virulence factors of K. pneumoniae during
the pathogenesis of infections. A paragraph on the frequency of
antimicrobial genes in K. pneumoniae has also been included, and the
research problem has been expanded and improved.

5. Why are siderophores described together with capsular virulence factors,
instead of grouping them with iron acquisition factors?

We are sorry for the confusion. The sizes of the tables have been
corrected and the different groups of virulence factors are now clearly
visible (Table 4 and 5). The wording of the description of the expression
of the virulence genes of strains lines 114-116 in the results section was
also improved).

6. Overall, methods used, and data presented is inconsistent - Introduction
presents in vitro model used, and Methods section also describes it, but
there is no mentioning it at all in the Results section. It should be clear,
which Methods were used to get each result.

We apologize for this inconsistency. We have corrected this
discrepancy throughout the text.

7. In all the cases, if the Authors claim the data was significant or
insignificant, the value given in the text is always "p<0.05". E.g. lines 72,
84...

Thanks for this observation. We have corrected this inconsistency in
the text.

8. As mentioned before, the Methods used to get Results are unclear. The
Authors present very briefly that in vitro tests were done, also regular PCR,
and q(rt)PCR was performed, but how the percentages of isolated
expressing virulence genes were calculated remains unclear. This must be
explained.

We appreciate your observation and kind comment. In the Methodology
section, we have clarified in detail and precision how the global
percentages of expression of the virulence genes in the K. pneumoniae
strains were obtained (section 4.7 Determination of K. pneumoniae
virulome expression using real-time PCR).

9. It is unclear, how the feature "hospital acquired" was attributed to the
isolates, the Authors should explain it.

Thanks for this observation. All patients with pneumonia (n=4) and
bacteremia (n=21) acquired these infections during their hospital stay,
after their admission for treating complications arising from other
comorbidities, such as diabetes, high blood pressure, chronic
obstructive pulmonary disease, and obesity. In the Methodology it has
been clarified with precision (lines 347-351).

10. What were the principles of separating the pulsotypes into separate
clades and groups, described in the text? I. e. why did the Authors chose to
describe the groups they selected, but not the bigger or smaller branches?

We appreciate your observation. The strains were grouped into clades
depending on the different patterns of PFGE (number of bands of the
bacterial chromosome), which were interpreted according to the criteria
previously described for the typing of bacterial strains [Tenover, F.C.;
Arbeit, R.D.; Goering, R.V.; Mickelsen, P,A.; Murray, B.E; Persing, D.H.;
Swaminathan, B.Interpreting chromosomal DNA restriction patterns
produced by pulsed-field gel electrophoresis: criteria for bacterial
strain typing. J. Clin. Microbiol. 1995, 33, 2233-2239]. This has been
clarified in the Methodology section 4.8 (lines 453-458). In the results
and discussion, we described the PFtypes belonging to the clades with
the largest number of strains, regardless of the size of the clade,
considering that these strains are the most predominant in hospitalacquired and community-acquired infections.

11. Also, all the pulsotype groups consist of a mix of hospital and community
acquired isolates. This raises a question, already mentioned in the remarks,
were the hospital acquired infections really nosocomial? This should be
discussed in the text.

Thanks for the observation, we have explained it more clearly in the
Methodology section 4.1 Origin of the strains (lines 347-351).

12. Lines 154-157 discusses information that was not presented in the
Results section.

Thanks for the kind comment, we have corrected this error.

13. In lines 217-219 the Authors present mortality data from a reference,
however, it is unclear, what information do the Authors have about the
destinies of the patients, from whom the K. pneumoniae used in this study
was collected.

Thank you for this comment. This study employed a cross-sectional
design and was conducted within a specific timeframe. During this
period, strains obtained from patients with hospital-acquired and
community-acquired infections were examined. Consequently, no
follow-up was conducted for hospitalized patients. Furthermore, it's
important to note that patient health information remained strictly
confidential within the hospital's policies and regulations.

14. Was the amplification efficiency of the primers used for qPCR
measured?

Thank you for the comment. We performed the amplification efficiency
of the primers from 1/10 serial dilutions of the cDNA of the strains
carrying the virulence genes (positive controls), of 1x103 and 1x104 in
5 μL volumes. The standard curve included 8 dilution points, each in
triplicate. The obtained Ct value was plotted against the base 10
logarithm of the DNA concentration and the curve points were fitted to
a straight line by linear regression. From the slope of the standard
curve, the efficiency of the amplification reaction was required using
the following formula:
Efficiency = 10 (-1/slope) –1.

15. It is unclear, how the expression data was calculated - what calculations
were done after Ct detection?

We appreciate your kind comment. In the Methodology section 4.7 (lines
428-435) we have clarified how the global percentages of expression of
the virulence genes in the K. pneumoniae strains were obtained.

16. In the Conclusions and in the text, the Authors present that "clones" of K.
pneumoniae could be detected. Is the presented PFGE data enough to claim
the presence of clones? Also, as long as it is unclear how hospital-acquired
infections were attributed to being hospital-acquired, the conclusions cannot
be considered.

Thank you for bringing this to our attention. You are correct, we have
removed the statement asserting the presence of clones in the K.
pneumoniae strains from the text. In relation to hospital-acquired
infections, we have provided clarity regarding their origin in the
Methodology section (4.1), which subsequently led to the establishment
of our conclusions.

Minor remarks:
1. In tables, the "Virulence gene expression no. (%)" and similar column
titles "no." is not the correct term.

Thanks for the observation. We have corrected it

2. The title of Table 2 is of wrong formatting and fused to the main text.

We have corrected it.

3. Pulsotypes term abbreviation is explained several times in the text, and
abbreviated differently. It should be made uniform.

We appreciate the observation. The term PFtypes has been corrected
uniformly throughout the text.

4. The information in Fig. 1 is barely readable - the genes found in the
isolates could be presented clearer using Heatmap or similar visualization.

We appreciate your suggestions. It was difficult to present such a large
number of strains in this figure. We acknowledge that the image's font
size appears to be small. We took great care to present a
comprehensive overview of the gels' direct output. Despite this, the
image resolution permits enlargement of each section and sufficient
magnification to facilitate detailed reading of the gene information.

5. The hemolysin gene name is incorrect in lines 180, 185

We have corrected it

6. In the Methods lines 256-257 the Authors present, that DNA content of
crudely lysed cells was measured by NanoDrop. What is the purpose and
reason of that...?

Thank you for your observation. We quantified the DNA concentration
to achieve an approximate dilution of all samples to the concentration
of 100 ng/μL, which was used for the endpoint PCR assays. This
approach aimed to ensure experimental consistency.

7. Line 256 should be -20 degrees, not ~20..?

Thank you. We have corrected it.

Reviewer 2 Report

The paper by Gloria Luz Paniagua-Contreras et al descibes the expression of the virulome related to antibiotic genotyping in nosocomial strains of Klebsiella pneumoniae. Taking into account understanding the wide spread of bacterial tolerance to antimicrobials, the understanding of the physiology of pathogens is an important challenge over the world.

The work is scientifically sounds and can be considered for publication. Nevertheless, some issues should be adressed.

Overall, while the experimental design is relevant, and idea is publication-worth, there are some issues to be adressed regarding data presentatio.

The introduction is too short and should be widened

Section 2.1 Please clarify the decision treshold for "expression".

"The most frequently detected antibiotic resistance genes among K. pneumoniae strains 

were beta-lactams (TEM, blaSHV, CITM, and CTXM1), tetracycline (tetA), sulfonamide 

(sul1), gentamicin (aac(3)-IV), streptomycin (aadA1), and trimethoprim (dfrA1).

Where this information comes? This work or from the literature? Do the strains used in the study carry ALL these genes?? It seems that each isolate should be analysed independently, and then the correlation of categorial data is required, i.e. positive/negative in virulence gene expression and the presence/absence of AR-gene.

Author Response

ID de manuscrito: ijms-2553055

Extensive expression of the virulome related to antibiotic genotyping in nosocomial strains of Klebsiella pneumoniae

Response letter
We are thankful to the reviewers for this opportunity to improve our
manuscript. We appreciate the insightful and constructive comments, and the
time spent on their review. In this revised version we addressed all review
comments and have modified the text accordingly. These additions,
clarifications and modifications have improved significantly the quality of the
manuscript. The following reply addresses these important issues point by
point. Our answers are written in bold script.

Reviewer comments:
Reviewer 2

1. The introduction is too short and should be widened

We appreciate your kind comment. We have expanded the introduction,
now describing with greater clarity and detail the importance of the
different virulence factors of K. pneumoniae during the pathogenesis of
infections. The research problem has also been expanded and
improved.

2. Section 2.1 Please clarify the decision treshold for "expression".

We appreciate your observation and kind comment. In the Methodology
section 4.7 (lines 428-435) we have clarified in detail the decision of the
Ct (threshold cycle) to determine the expression of virulence genes in
K. pneumoniae strains.

3. "The most frequently detected antibiotic resistance genes among K.
pneumoniae strains were beta-lactams (TEM, blaSHV, CITM, and CTXM1),
tetracycline (tetA), sulfonamide (sul1), gentamicin (aac(3)-IV), streptomycin
(aadA1), and trimethoprim (dfrA1).
Where this information comes? This work or from the literature? Do the
strains used in the study carry ALL these genes?? It seems that each isolate
should be analysed independently, and then the correlation of categorial data
is required, i.e. positive/negative in virulence gene expression and the
presence/absence of AR-gene

Thanks for your comment. All percentages of genes encoding antibiotic
resistance were obtained in this study from K. pneumoniae strains
isolated from patients with hospital-acquired and community-acquired
infections.
Regarding the expression of virulence genes, we apologize for the
confusion. In a previous study we evaluated and published the
prevalence of virulence genes in these same strains of K. pneumoniae
(Bautista-Cerón A, Monroy-Pérez E, García-Cortés LR, Rojas-Jiménez
EA, Vaca-Paniagua F, Paniagua-Contreras GL. Hypervirulence and
Multiresistance to Antibiotics in Klebsiella pneumoniae Strains isolated
from patients with hospital-acquired and community-acquired
infections in a Mexican medical center. Microorganisms.
2022;10(10):2043), so in this new study we focused solely on
establishing the frequency of virulence gene expression and its
relationship to antibiotic resistance genotype after infection in an in
vitro model in human epithelial cell lines. To improve clarity, we have
further refined sections 4.1 and Methodology 4.7 to provide full details
on the origin of K. pneumoniae strains from different clinical isolates
and how we used previously characterized samples as positive controls
for real-time PCR assays (section 4.7).

Round 2

Reviewer 1 Report

I want to thank the Authors for addressing part of my concerns with the manuscript. However, some issues were not resolved, and the understanding of the experiments performed and data described can still be confusing for the reader. 

In particular:

1. I have mentioned in previous review, that it should be clear, which Methods were used to get each result. To my mind, the Results section has not been clarified enough. It is still hard to understand which method - regular PCR or in vitro assay+qPCR was used for which genes, and what was the motivation behind using one or the other.

2. As far as I understood, regular PCR was used to detect resistance genes, and in vitro assay+q(rt)PCR - for virulence genes. What about virulence genes present in the genome, but not expressed? No information is given on that; however, this could be an important piece of information.

2. Mistakes are still left in the texts - not all beta lactamase names are corrected, p=<0.05 can still be found in the place where statistically insignificant results are presented.

3. The Authors supply an answer to the question 14 (Was the amplification efficiency of the primers used for qPCR measured?), that yes, it was. However, what was the efficiency of the primers, was it sufficient to make qPCR results reliable?

4. The Authors explained their choice to leave Fig 1 as is, which is their right, however, my concern, is that the important information presented there as the gene expression patterns is almost unusable.

5. I would still argue, that using NanoDrop to measure and evaluate DNA concentration on crude lysates is not an appropriate technique.

6. The in vitro experiment is still presented very minimally - e.g. what was the logic in choosing cell lines, planning the experimental setting.

7. How was the phenotypic resistance data correlated with antibiotic resistance genes that were detected by molecular methods?

English is mostly clear and understandable, however there are minor mistakes and grammar mistakes that must be fixed in the final manuscrip. Perhaps language editors could do that.

Author Response

ID de manuscrito: ijms-2553055

Extensive expression of the virulome related to antibiotic genotyping in nosocomial strains of Klebsiella pneumoniae

Response letter

We are thankful to the reviewers for this opportunity to improve our manuscript. We appreciate the insightful and constructive comments, and the time spent on their review. In this second revised version, we addressed all review comments and have modified the text accordingly. These additions, clarifications and modifications have significantly improved the quality of the manuscript. The following reply addresses these important issues point by point. Our answers are written in bold script.

Reviewer comments:

Reviewer 1

In particular:

1. I have mentioned in previous review, that it should be clear, which Methods were used to get each result. To my mind, the Results section has not been clarified enough. It is still hard to understand which method - regular PCR or in vitro assay+qPCR was used for which genes, and what was the motivation behind using one or the other.

We appreciate the observation. Now in the results section (items 2.1, 2.2, 2.3, and 2.4) we have described in detail and fully clarified the PCR methods used to detect antibiotic resistance genes and real-time PCR (qPCR) for the determination of expression of virulence genes in K. pneumoniae strains.

2. As far as I understood, regular PCR was used to detect resistance genes, and in vitro assay +q(rt)PCR - for virulence genes. What about virulence genes present in the genome, but not expressed? No information is given on that; however, this could be an important piece of information.

Thank you for the comment. We have addressed this issue in the discussion section within lines 374-379.

2. Mistakes are still left in the texts - not all beta lactamase names are corrected, p=<0.05 can still be found in the place where statistically insignificant results are presented.

We apologize for these errors. We made the corrections throughout the manuscript.

3. The Authors supply an answer to the question 14 (Was the amplification efficiency of the primers used for qPCR measured?), that yes, it was. However, what was the efficiency of the primers, was it sufficient to make qPCR results reliable?

Thanks for the comment. To determine the efficiency of the primers, we carried out for each pair of primers a curve of 8 dilutions in triplicate of 1x103 and 1x10-1 with the DNA of the strains carrying each gene (positive controls). At the end of the assay, the Rotor-Gene Q 5plex HRM System software version 2.3.1.49 (Qiagen) automatically calculated the threshold cycle (TC) and the R2 value, which was on average 0.999. The amplification efficiency in the linear range of the assay was on average 95%, which demonstrates a high and homogeneous amplification efficiency of all the primers and is sufficient for the qPCR results to be reliable.

4. The Authors explained their choice to leave Fig 1 as is, which is their right, however, my concern, is that the important information presented there as the gene expression patterns is almost unusable.

We are sensitive to your observation. Therefore, we have created a new figure in the form of a heatmap (figure 2), where the unsupervised hierarchical grouping of K. pneumoniae strains is analyzed globally and systematically. This allowed us to present the different clades of the strains according to their profile, expression of the virulome associated with the genotype-phenotype of antibiotic resistance, and clinical origin. We provide a detailed analysis of this new figure in the results section (item 4.9) and further discussed its implications in the discussion section. We consider that this figure serves as a detailed complement of the information in Figure 1 (PFGE) where the different virulome expression profiles related to the PFtype are presented. We consider that this new figure answers thoughtfully this specific and important point.

5. I would still argue, that using NanoDrop to measure and evaluate DNA concentration on crude lysates is not an appropriate technique.

We appreciate the reviewer's point and fully acknowledge the limitations associated with employing NanoDrop for quantifying DNA concentration in crude lysates. We recognize that this method may not

yield the quantitative precision of purified DNA measurements. However, for qualitative assessment of genes (present or absent) we consider that this technique offers enough reliability. It's important to note that our study followed a strict adherence to a systematic protocol across all samples, which ensured uniformity in sample treatment, allowing for meaningful comparative analysis despite these limitations.

6. The in vitro experiment is still presented very minimally - e.g. what was the logic in choosing cell lines, planning the experimental setting.

We appreciate your observation. The preparation of the cell lines, as well as the objective of using them in this research, has been described and expanded in detail in the material and methods section (section 4.5).

7. How was the phenotypic resistance data correlated with antibiotic resistance genes that were detected by molecular methods?

We have now correlated the genotype-phenotype of antibiotic resistance in the new figure (figure 2) of the unsupervised hierarchical clustering, and these findings have been described in the results section 2.6.

Reviewer 2 Report

While Authors submitted the rebuttle letter on Spanish and the reviewer was forced to use the online translator to read an answer, the paper now has been imroved and became readable. Therefore it can be considered for publication. 

Author Response

ID de manuscrito: ijms-2553055

Extensive expression of the virulome related to antibiotic genotyping in nosocomial strains of Klebsiella pneumoniae

Response letter

We thank the Reviewer for the Insightful and Constructive Comments, and the time dedicated to the review of our manuscript. These additions, clarifications and modifications have improved significantly the quality of the manuscript.

Round 3

Reviewer 1 Report

The Authors have improved the manuscript according my last comments, many questions were answered and text clarified. However, some remarks still remain unanswered:

1. I have mentioned in previous review, that it should be clear, which Methods were used to get each result. To my mind, the Results section has not been clarified enough

--- The use of Methods was clarified, however, the choice to use one or another remains unclear to me. E.g. why virulence genes couldn’t be detected by regular PCR? Or, why ARGs couldn’t be detected by qPCR? Why the separation of methods?

2. As far as I understood, regular PCR was used to detect resistance genes, and in vitro assay +q(rt)PCR - for virulence genes. What about virulence genes present in the genome, but not expressed? No information is given on that; however, this could be an important piece of information. 

--- The lines in the discussion section (374-379) do not answer to the issue raised by this question.

English is fine.

Author Response

ID de manuscrito: ijms-2553055

Extensive expression of the virulome related to antibiotic genotyping in nosocomial strains of Klebsiella pneumoniae

Response letter

We are thankful to the reviewer for this opportunity to improve our manuscript. We appreciate the insightful and constructive comments, and the time spent on their review. In this third revised version we addressed all review comments and have modified the text accordingly. These additions, clarifications and modifications have improved significantly the quality of the manuscript. The following reply addresses these important issues point by point. Our answers are written in bold script.

Reviewer comments:

Reviewer 1

Comments and Suggestions for Authors

The Authors have improved the manuscript according my last comments, many questions were answered and text clarified. However, some remarks still remain unanswered:

1. I have mentioned in previous review, that it should be clear, which Methods were used to get each result. To my mind, the Results section has not been clarified enough

We appreciate this observation. In an effort to enhance the clarity of both the Methods and Results sections, we have improved the description of our procedures. Specifically, we have made significant additions in Section 4.2, for a more detailed description of the DNA extraction method. In Section 4.3, we have provided a more comprehensive information of the identification of K. pneumoniae using conventional PCR. Furthermore, in Section 4.5, we have expanded upon the detection of antibiotic resistance genes via conventional PCR, ensuring that the methodology is thoroughly explained. We believe these enhancements will contribute to a better understanding of our research methodology.

We have incorporated important additions to our manuscript. A new section, Section 4.6, has been introduced to describe the preparation of bacterial dilutions for the in vitro infection model. Additionally, we have dedicated

another section, Section 4.7, to describe the procedures of human epithelial cell lines cultures.

To further enhance the comprehensibility of our methodology, we have made refinements in Section 4.8, where we now provide a comprehensive account of RNA extraction and t reverse transcription into cDNA. In Section 4.9, we have improved the description of the methodology used for the determination of virulome expression through real-time PCR. We trust that these enhancements will significantly contribute to the overall clarity of the methods used.

With respect to the results section, and with the purpose of clarifying and improving their understanding, we have expanded and described with greater clarity sections 2.1 (Multidrug-resistance phenotype), 2.2 (Origin of the strains and in vitro infection of human epithelial cell lines to determine virulome expression) and 2.3 (Detection of antibiotic resistance genes by conventional PCR). We consider that these new modifications and clarifications enhance the clarity of the global results described in the 5 tables and in the two figures.

--- The use of Methods was clarified, however, the choice to use one or another remains unclear to me. E.g. why virulence genes couldn’t be detected by regular PCR? Or, why ARGs couldn’t be detected by qPCR? Why the separation of methods?

Certainly, we appreciate your attention to our methodology. We want to highlight that, indeed, virulence genes can be detected using conventional PCR. In fact our working group has already determined and recently published the frequency of virulence genes by the conventional PCR method related to hypermucoviscosity and capsular serotypes in the genome of these K. pneumoniae strains (Bautista-Cerón, A.; Monroy-Pérez, E.; García-Cortés, L.R.; Rojas-Jiménez, E.A.; Vaca-Paniagua, F.; Paniagua-Contreras, G.L. Hypervirulence and Multiresistance to Antibiotics in Klebsiella pneumoniae Strains Isolated from Patients with Hospital- and Community-Acquired Infections in a Mexican Medical Center. Microorganisms. 2022, 10, 2043), which has been included in a paragraph in the discussion (lines 293-300), however, as a continuation of our research, in this new work we now implemented an in vitro infection model using human cell lines with K. pneumoniae strains to establish the expression of virulence genes by real-time PCR. Therefore, the real-time PCR method using the Rotor Gene (QIAGEN) equipment allowed us to detect and quantify accurately the precise values of the CT (threshold cycle) of each strain based on a standard curve carried out with positive controls (strains carrying the genes) and with internal controls (Housekeeping), which would not be possible using the conventional PCR method.

Regarding the antibiotic resistance genotype (AGR), the objective was not to determine the expression of these genes, but rather to exclusively determine qualitatively the presence or absence of these genes in the genome of the strains, so it was more practical and simple to use the conventional PCR method.

2. As far as I understood, regular PCR was used to detect resistance genes, and in vitro assay +q(rt)PCR - for virulence genes. What about virulence genes present in the genome, but not expressed? No information is given on that; however, this could be an important piece of information.

--- The lines in the discussion section (374-379) do not answer to the issue raised by this question.

Thank you for this observation and we apologize for not having thoroughly answered the question the previous lines (374-379) regarding what happens to the virulence genes present in the genome, but not expressed.
We now in the discussion section (now lines 375-390) have clarified and more broadly discussed our hypotheses as to why the virulence genes present in the genome were probably not expressed during the in vitro model of infection of the cell lines with the strains. of K. pneumoniae.
